# A Comparative Analysis of Characteristics and Synoptic Backgrounds of Extreme Heat Events over Two Urban Agglomerations in Southeast China

Xiaoyan Sun [1,2], Xiaoyu Gao [2,*], Yali Luo [1,2,*], Wai-Kin Wong [3] and Haiming Xu [1]

1   Collaborative Innovation Center on Forecast and Evaluation of Meteorological Disasters, Key Laboratory of Meteorological Disasters of Ministry of Education, Nanjing University of Information Science and Technology, Nanjing 210044, China
2   State Key Laboratory of Severe Weather, Chinese Academy of Meteorological Sciences, Beijing 100081, China
3   Hong Kong Observatory, Hong Kong 999077, China
*   Correspondence: gaoxy@cma.gov.cn (X.G.); ylluo@cma.gov.cn (Y.L.)

**Abstract:** Based on high-resolution surface observation and reanalysis data, this paper analyzes the extreme heat events (EHEs) over two densely populated urban agglomerations in southeast China, namely the Yangtze River Delta (YRD) and the Pearl River Delta (PRD), including the spatial–temporal distribution of heatwaves and warm nights and the synoptic backgrounds for regional heatwaves. The results show that the occurrence frequency of EHEs is modulated significantly by local underlying features (i.e., land–sea contrast, terrain), and the strong nocturnal urban heat island effects make warm nights much more likely to occur in cities than rural areas during heatwaves. About 80% of the YRD regional heatwaves occur from 15 July to 15 August, while a lower fraction (53%) of the PRD heatwaves is found during this mid-summer period, which partially explains the warm-season average intensity of the former being 2–3 times the latter. A persistent, profound subtropical high is the dominant synoptic system responsible for the mid-summer YRD heatwaves, which forces significant descending motion leading to long-duration sunny weather. The mid-summer PRD heatwaves involve both high-pressure systems and tropical cyclones (TCs). A TC is present to the east of the PRD region on most (about 72%) PRD heatwave days. The organized northerly winds in the planetary boundary layer in the outer circulation of the TC transport the inland warm air, which is heated by the foehn effect at the lee side of the Nanling Mountains and possibly also the surface sensible heat flux, towards the PRD region, leading to the occurrence of the extremely high temperatures.

**Keywords:** extreme heat event; urban heat island; subtropical high; tropical cyclone





## 1. Introduction

Extreme heat events (EHEs) seriously threaten human health [1], the economy [2], and the ecosystem [3]. Their frequency and duration have gone through a dramatic increase worldwide during the past 20 years [4] and are projected to further increase in the scenario of continuing global warming [5]. It is almost certain that social issues and health-related risks induced by EHEs will bring even greater challenges to governments in the near future.

Previous studies suggest that EHEs could result from the coaction of multiple physical processes at various spatiotemporal scales, including climate variability, synoptic weather systems, and local circulations associated with the topography and inhomogeneous conditions of land surface [6]. For example, El Niño could cause extensive EHEs worldwide, especially during boreal winter, while positive phases of the North Pacific index could lead to EHEs in the southeast United States and East Asia [7]. A persistent anticyclone system often acted as the main driver of EHEs over many regions [8,9]. At higher latitudes, the blocking highs obstructed the invasion of cold air, which were responsible for the Chicago

heatwave in 1995 [10], the European heatwave in 2003 [11,12], and the Russian heatwave in 2010 [13]. At mid-latitudes, the stationary subtropical highs could induce strong subsidence and dry air conditions over southeast China [14] and transport hot and dry air from the inland desert areas of central Australia to the southern part of the continent [15], favoring higher temperatures and the occurrence of EHEs. Moreover, tropical cyclones (TCs) could induce sudden and dramatic temperature rise and short-term EHEs under certain circumstances [16–18], mainly due to the subsidence-induced adiabatic heating at the periphery of the TCs' spiral rainbands. The TC circulation could also cooperate with local topography, producing the foehn effect, which causes sudden downslope heating [16]. In addition, a dry land surface usually releases more sensible heat (instead of latent heat) into the atmosphere [19] and can contribute to the occurrence of extremely high temperatures [20–22]. Substantial changes in the heat capacity and moisture of land surface due to urbanization lead to the urban heat island (UHI) effects, which could have a profound impact on EHEs over cities [23].

Over the past three decades, China has undergone rapid economic development, leading to dramatic growth in urban areas and population, especially in southeast China [24]. Two urban agglomerations, the YRD and the Pearl River Delta (PRD), are the most densely populated and flourishing economic areas in East Asia (see Figure 1 for their locations). The YRD, including the megacities of Shanghai, Nanjing, Suzhou, and Hangzhou, is an important intersection of the Belt and Road and the Yangtze River economic belt. The PRD, including the megacities of Guangzhou, Hong Kong, and Macao, is ranked as one of the four great bays in the world with diversified industries and a prosperous economy. As shown in Figure 1b,c, the underlying surface over PRD and YRD exhibits complex features with different shapes of urban agglomerations adjacent to mountains and seas. With low latitudes and wet climates, YRD and PRD experience high temperatures in boreal summer that are exacerbated by the UHI effect [25]. The YRD summer-time warming from 1979 to 2008 primarily results from the significant increase in maximum temperature, with the UHI effect accounting for 36–68% of the total regional warming [26]. Urbanization contributes to more than one-third of the increase in the intensity of EHEs in the YRD region from 1971 to 2013, which is comparable to the contribution of greenhouse gases [27]. A case study indicates that the maximum UHI intensity was up to 2.2 °C in Suzhou city during a hot weather episode from 25 July to 1 August 2007, which is much greater than the 37-year (from 1970 to 2006) summer average of 0.35 °C [28]. Considering the numerous people and infrastructures influenced by extreme temperatures, a better understanding of the characteristics and driving mechanisms of EHEs over the two regions would help in the warning and mitigation of the adverse effects of EHEs.

Numerous studies have investigated the factors and mechanisms governing the occurrence of EHEs over eastern China. Ding et al. [29] found that the occurrence frequency of heatwaves in most regions of China increased during 1961–2007, which was closely related to the changes in rainy days and atmospheric circulation patterns. Chen et al. [30] found that a strong northward water vapor flux reduced the nocturnal longwave radiation cooling around Beijing and was associated with warm nights in Beijing during July and August. Chen et al. [31] found that the warming of the western tropical Pacific forced an abnormally intensified subtropical high, which was responsible for the 2017 extremely hot mid-summer in Central and South China. Zhong et al. [32] found that frequent TC activities could indirectly lead to more hot days along the lower and middle reaches of the Yangtze River in East China, regardless of the cooling effects directly induced by TCs. Li et al. [33] showed that a persistent anticyclonic anomaly with a quasi-barotropic structure is a prerequisite for the compound heat extremes (i.e., the continuous occurrences of hot days and warm nights) in the middle and lower reaches of the Yangtze River. While the middle and lower reaches of the Yangtze River experience the highest frequency of EHEs in eastern China [29], only limited studies have focused on EHEs over the urban agglomerations, the PRD region in South China in particular.

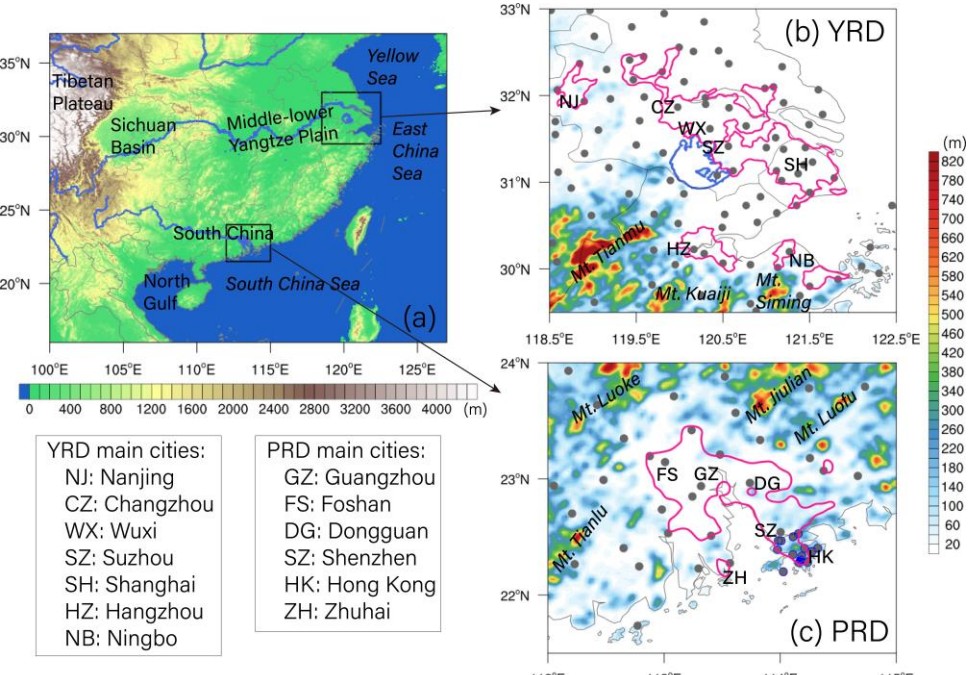

**Figure 1.** Topography (color shadings) over (**a**) southern China and surrounding areas and the (**b**) YRD and (**c**) PRD regions, respectively. In (**b**,**c**), gray dots denote the surface weather stations with one blue dot denoting the Hong Kong Observatory (HKO) station; pink lines denote the urban regions; names of major cities are given by abbreviation; major mountains are labeled. Blue lines in (**b**) indicate Lake Taihu. Gray lines indicate the national and provincial boundaries and coastal lines.

To fill this gap of knowledge, this study aims to analyze and compare the spatial–temporal distribution and synoptic backgrounds of EHEs over the YRD and PRD regions. Our analysis focuses on the warm seasons (May to August) in a ten-year period (2008–2017) with well-established urban agglomerations in both regions, as the rapid urbanization over the two regions started in the early to middle 1990s. The remainder of the paper is organized as follows: data and methods are introduced in Section 2; spatial–temporal distribution and synoptic backgrounds of EHEs are discussed in Sections 3 and 4, respectively; and a summary and conclusions are provided in Section 5.

## 2. Data and Methods

### 2.1. Data

The daily maximum and minimum 2 m air temperature (denoted as $T_{max}$ and $T_{min}$) observed at the surface weather stations (dots in Figure 1b,c) is used to identify the EHEs. There are 93 and 46 stations in the YRD and PRD regions, respectively, including 14 stations in Hong Kong. This dataset has been quality controlled by the National Meteorological Information Center (NMIC) of China and widely used in climate studies [25,34–36]. The 5th generation of ECMWF Reanalysis (ERA5; available every hour on a $0.25° \times 0.25°$ grid and 37 vertical levels) are used to analyze the synoptic backgrounds. The TC best track dataset [37,38] provided by the China Meteorological Administration (CMA) is used to identify TCs related to the heatwaves. Moreover, the urban and rural regions are distinguished based on the fourth version of composite satellite-based nighttime lights data gathered in 2013 derived from the Defense Meteorological Satellite Program's Operational Linescan System (DMSP/OLS) of the United States (https://ngdc.noaa.gov/eog/dmsp/downloadV4composites.html, accessed on 10 June 2022). The digital numbers (DN) value of 57 is used as the urban threshold based on Wang et al. [39].

*2.2. Definition and Analysis of EHEs*

This study focuses on two types of EHEs, namely, heatwaves and their accompanying warm nights. According to the Glossary of Meteorology of the American Meteorological Society (AMS) [40], a heatwave is defined as a period of abnormally and uncomfortably hot and usually humid weather. The identification of heatwaves in literature involves $T_{max}$ above a threshold during continuous days. For example, Schoetter et al. [41] focused on periods of over three consecutive days above the 98th percentile of $T_{max}$. The indices used to characterize heatwaves vary among different studies [10,12,41,42], which makes it difficult to quantitatively compare their results. In this study, we use a unified metric for both the YRD and PRD regions. Specifically, a period of over three consecutive days with $T_{max} > 35\ °C$ is identified as a heatwave, following CMA's stipulation. This definition of heatwaves is similar to that of the WMO with a lower fixed threshold of 32 °C [40]. For each weather station, we use two indices to discuss the characteristics of heatwaves, i.e., the occurrence frequency and the average cumulative intensity. The occurrence frequency of heatwave at a station is the total number of heatwaves that occurred at the station during the 10 warm seasons. The cumulative intensity of each heatwave at a station is the sum of $T_{max}$ beyond 35 °C (i.e., $T_{max}$—35 °C) on each day during the heatwave, which is averaged among all the heatwaves to get the average cumulative intensity at the station. Moreover, we define a regional heatwave as a period with at least half of the stations in the region (YRD or PRD) simultaneously experiencing a heatwave. Note that, among the 14 stations in Hong Kong, only the Hong Kong Observatory station is used to analyze the regional heatwaves, because the stations in Hong Kong are distributed much more densely than other stations in the PRD (Figure 1c).

The possible occurrences of warm nights in heatwaves can aggravate the adverse impacts of heatwaves due to the prolonged high temperature in the day and night [43]. Such compound hot extremes with continuous occurrences of hot days and warm nights are recently noticed to be more frequent and intense over cities than surrounding rural areas, due to the UHI effect [44]. To define the compound EHEs, a metric is needed for warm nights. As the threshold used to define EHEs (35 °C) is about the 90th percentile of $T_{max}$ in both the YRD and PRD, we similarly use the 90th percentile of $T_{min}$ (approximately 27 °C) to define a warm night, i.e., a night with $T_{min} > 27\ °C$. For the compound EHEs, i.e., heatwaves with warm nights, the proportion and the cumulative intensity of warm nights are calculated. The proportion is the ratio of total number of warm nights to the total number of days in a heatwave event, and the cumulative intensity of warm nights is the sum of $T_{min}$—27 °C during a heatwave event.

## 3. Characteristics of EHEs over the Two Regions

*3.1. Spatial Distribution of EHEs*

Substantial spatial variations of EHEs are observed over the PRD and YRD regions. In the YRD region, the occurrence frequency of heatwaves largely decreases from the mountains in the southwest to the city belt, and then to its northeast (Figure 2a). More than 30 heatwaves are observed at most mountain stations and the cities of Hangzhou and Ningbo adjacent to the mountains. The number of heatwaves reduces to 20–30 at most stations in the belt-shaped urban area. The 37 urban stations in the YRD region experience an average of 23.6 heatwaves in the 10 warm seasons, and the highest frequency of 42 is recorded in the capital city of Hangzhou. At the stations north of the city belt, the heatwave frequencies are lower than 20, likely due to the cold advection of sea breeze over the open flat land. Although much fewer heatwaves occur in the city belt than in the mountains, their intensities are roughly comparable between the two areas, mostly with a cumulative intensity of >13 °C (Figure 2b).

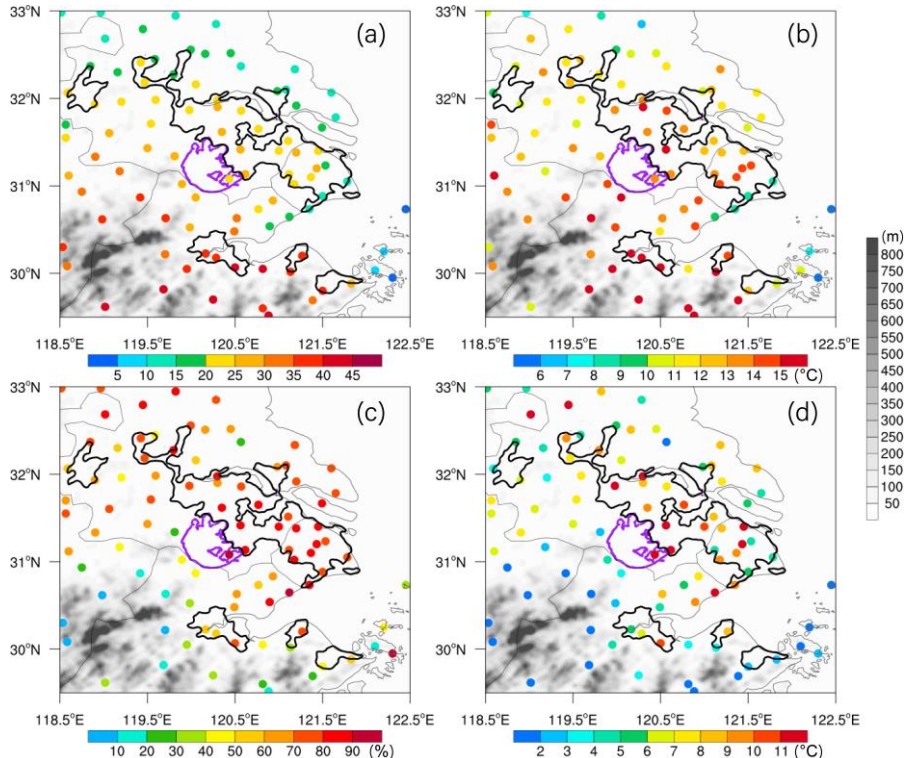

**Figure 2.** Spatial distribution of EHEs occurred over the YRD region in the warm seasons during 2008–2017. (**a**) Occurrence frequency of heatwaves. (**b**) Average cumulative intensity of heatwaves. (**c**) Proportion of warm nights in heatwaves. (**d**) Average cumulative intensity of warm nights. Gray shadings denote the topography. Black lines indicate the urban regions, and thin gray lines denote the provincial boundaries and coastlines. Purple lines show Lake Taihu.

The spatial distributions of warm nights during heatwaves in the YRD region exhibit a pattern that highlights the flat lands, especially the city belt (Figure 2c,d). For most urban stations, more than 70% of heatwave days are accompanied by warm nights, while this proportion is less than 40% in the mountains (Figure 2c). Meanwhile, the intensity of warm nights in the cities is much higher than that in the mountains (mostly >8 °C vs. <3 °C; Figure 2d). The proportion and cumulative intensity of warm nights averaged over the YRD urban stations are about 73% and 7.8 °C, while only 53% and 4.8 °C for the rural stations.

The heatwaves over the PRD region exhibit a clear contrast between the coastal and inland areas (Figure 3a,b). The heatwaves rarely occur (mostly <15) in the coastal area (i.e., to the south of the red line in Figure 3a, which is oriented roughly in parallel to the coastline), but sharply increase to >30 in the northern inland area. The heatwaves in the coastal area are also relatively weaker than the inland ones (the average cumulative intensity of mostly <3 °C vs. >4.5 °C). Compared to the YRD region, the intensity of heatwaves over PRD is much lower (c.f. Figures 2b and 3b), with the highest average cumulative intensity in Guangzhou city of only 5–6 °C. This is closely related to the earlier occurrences of heatwaves in June over PRD (Section 3.2) and the location of Western North Pacific Subtropical High (WNPSH) in summer (Section 4).

Similar to the YRD region, the cities in the PRD (including the coastal cities such as Hong Kong and Zhuhai) experience much larger fractions of warm nights in heatwaves than the rural areas (Figure 3c,d). The average proportion and cumulative intensity are 66% and 3.5 °C for the urban stations, while only 31% and 1.0 °C for the rural stations. These results indicate that most heatwaves in the rural areas turn into cool nights after sunset, while the nocturnal temperature remains high in the cities. These results are qualitatively consistent with the previous finding of significantly larger increasing trends of compound hot extremes than independent hot days or independent warm nights in

cities over southeast China during the rapid urbanization period from the mid-1990s to 2020 [44]. The building materials of urban canopies have larger thermal conductivity and heat capacity, leading to a larger heat storage than rural areas. This slows down the nocturnal cooling and makes significant UHI effects at night [45].

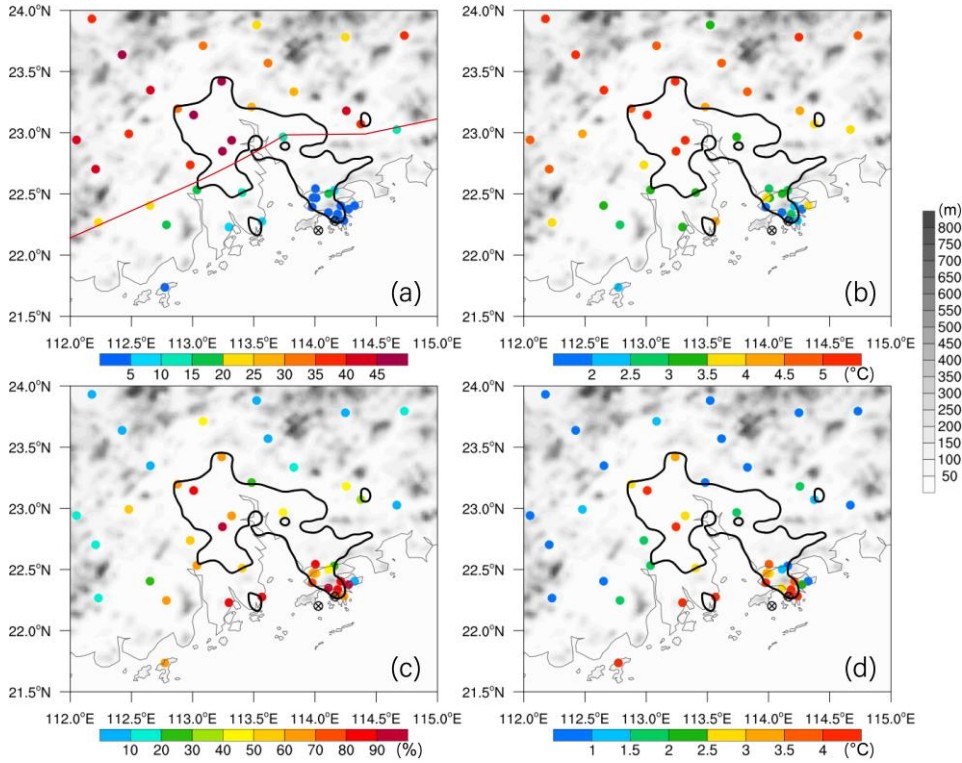

**Figure 3.** Spatial distribution of EHEs occurred over the PRD region in the warm seasons during 2008–2017. (**a**) Occurrence frequency of heatwaves. (**b**) Average cumulative intensity of heatwaves. (**c**) Proportion of warm nights in heatwaves. (**d**) Average cumulative intensity of warm nights. Gray shadings denote the topography. Black lines indicate the urban regions, and thin gray lines denote the provincial boundaries and coastlines. Crossed circles indicate that no heatwave occurred. The red line in (**a**) roughly separates the coastal and inland areas.

In summary, local geographical features substantially influence the occurrences of heatwaves and the impact of the UHI effect on EHEs is modulated by the presence of water and mountains over the PRD and YRD regions. The cities adjacent to inland mountains (e.g., Hangzhou in the YRD and Guangzhou in the PRD) experience the most frequent and intense heatwaves, while those adjacent to the open seas (e.g., Hong Kong and eastern Shanghai) show less frequent heatwaves with weaker intensity (Figures 2a and 3a). The urban and rural stations sharing similar geography observe smaller differences in the heatwave frequency (Figures 2a and 3a), but both inland and coastal cities experience much more warm nights in heatwaves (Figures 2c and 3c) with substantially higher intensity (Figures 2d and 3d) than the rural areas. The result emphasizes the important role of the UHI effect in producing nocturnal high temperatures in both regions. Moreover, the intensities of heatwaves and the accompanied warm nights over the YRD are about 2–3 times higher than those in the PRD (c.f., Figures 2b,d and 3b,d), which can be attributed to the differences in the dates and synoptic backgrounds of the heatwaves between the two regions to be shown in the following sections.

### 3.2. Temporal Variation of the Regional Heatwaves

In total, 137 days of the YRD regional heatwave and 102 days of the PRD regional heatwave are found during the 10 warm seasons, and all of them occur in summer (JJA).

The YRD regional heatwaves begin on 1 July and end on 23 August, and the PRD regional heatwaves begin on 1 June and end on 29 August (Figure 4a). It follows that occurrences of the heatwaves in the YRD region are more concentrated in mid-summer, while those in the PRD occur also in June and near the end of August. About 80% of the YRD regional heatwave days and 53% of the PRD regional heatwave days are from 15 July to 15 August (referred to as 'mid-summer' hereafter). Only 13.1% and 7.3% of the YRD regional heatwave days occur before and after the mid-summer, respectively, while the proportions in the PRD are nearly doubled or more (33.3% and 13.7%). The PRD is located in lower latitudes than the YRD, which could largely explain the earlier and later occurrences of heatwaves in the PRD. In June, South China is at the late stage of its early-summer rainy season [46] and under the influence of the WNPSH, while the Yangtze River basin is out of the reach of WNPSH. Thus, heatwaves could strike the PRD but hardly influence YRD in June. In mid-summer, the WNPSH often covers the middle and lower reaches of the Yangtze River, favoring the formation of the YRD heatwaves, while the PRD is to the southwest of WNPSH and tropical systems occasionally influence southeast China. These indicate that the relevant synoptic backgrounds of heatwaves over the YRD and PRD in mid-summer might be different. The following section will focus on synoptic patterns governing the YRD and PRD regional heatwaves during mid-summer.

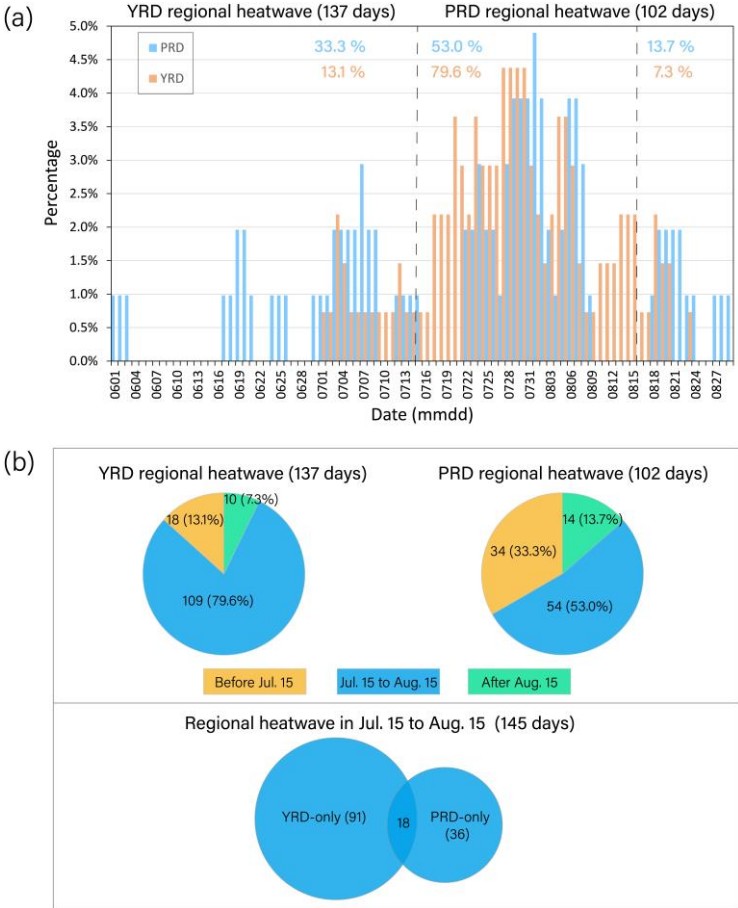

**Figure 4.** Occurrence frequency of regional heatwaves over the PRD and YRD regions, respectively, during the 10 warm seasons. (**a**) Fraction of heatwaves on each day to the total heatwave days during the warm season. (**b**) Numbers (fractions; unit: %) of the regional heatwave days in the entire warm season and a variety of subperiods, respectively.

## 4. Synoptic Backgrounds for the Mid-Summer YRD- and PRD-Regional Heatwaves

In mid-summer, 109 and 54 regional heatwave days are found over the YRD and PRD regions, respectively, including 91 days of the YRD-only heatwave, 36 days of the PRD-only

heatwave, and 18 days of simultaneous heatwaves over both regions. Synoptic patterns are composited for each of the three groups and their subgroups. Specifically, the geopotential height (GPH), total cloud cover, net solar radiation at the surface, horizontal wind vector, and vertical wind velocity available hourly in the ERA5 dataset are averaged to produce their daily values. Then, for each group and/or subgroup of the regional heatwaves, all the daily values are averaged to show their composite synoptic backgrounds. The anomalies are defined as the differences between the heatwave days and mid-summer. Anomalies of geopotential height shown in the following figures have passed the Student t-test of 95% significance.

### 4.1. The YRD-Only Heatwaves

At 300 hPa, the YRD region is situated in an extensive, east–west-oriented, high-pressure area with GPH > 9740 gpm, downward motion, and low horizontal wind speeds (Figure 5a). The positive anomaly of geopotential height over and to the north of the YRD is accompanied by easterly anomalies (Figure 5d), reflecting that the upper-level westerly jet in mid-summer is weakened during the YRD regional heatwaves.

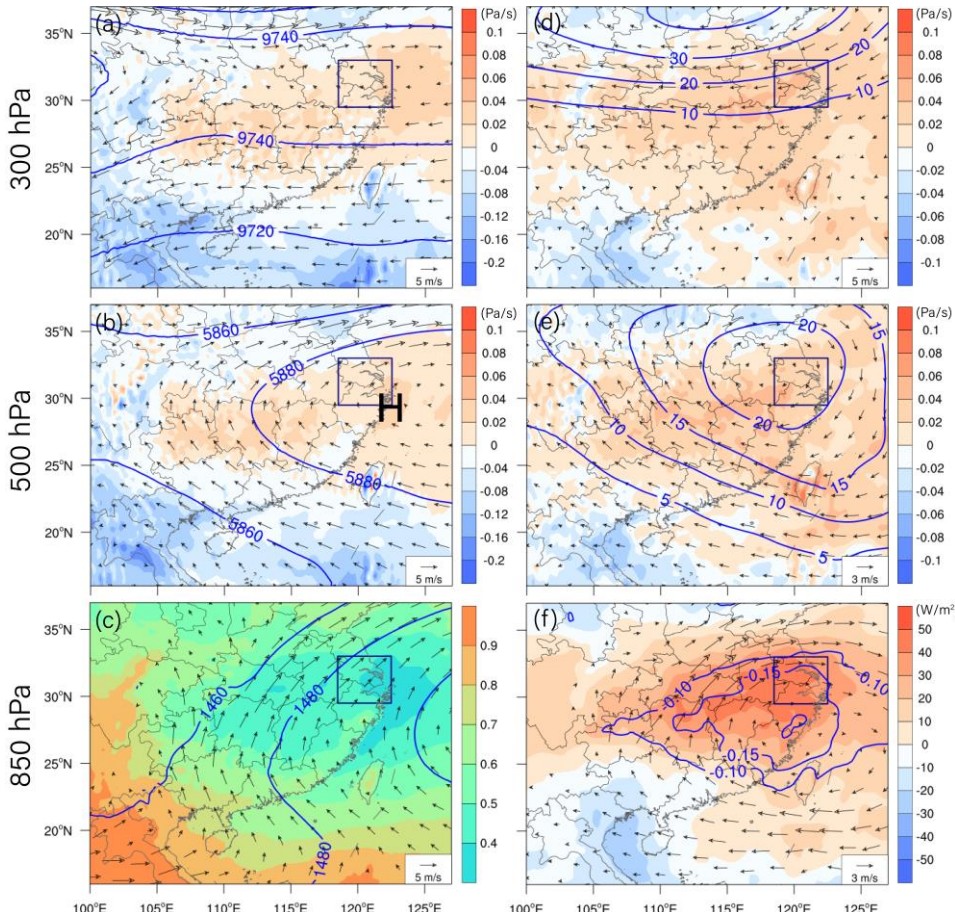

**Figure 5.** Synoptic backgrounds for the YRD-only heatwaves. (**a**) The 300 hPa geopotential height (blue contours; unit: gpm), horizontal wind vector (arrows), and vertical velocity (shadings). (**b**) Same as (**a**), but at 500 hPa. (**c**) Geopotential height (blue contours; unit: gpm) and horizontal wind vector (arrows) at 850 hPa, and total cloud cover (shadings). (**d**) Anomalies of geopotential height (blue contours; unit: gpm), horizontal wind vector (arrows), and vertical velocity (shadings) at 300 hPa. (**e**) Same as (**d**) but at 500 hPa. (**f**) Anomalies of total cloud cover (blue contours), 850 hPa horizontal wind vectors (arrows), and net solar radiation at the surface (shadings; positive downward). Navy rectangle denotes the YRD region. Gray lines indicate the national and provincial boundaries and coastal lines.

In the middle troposphere (500 hPa), the strong WNPSH extends westward to cover the middle and lower reaches of the Yangtze River (Figure 5b), accompanied by descending motion over the region (Figure 5a,b) with large anomalies of both GPH and vertical velocity (Figure 5d,e).

The lower troposphere (850 hPa) also experiences a strong high pressure extending from the North Pacific with GPH > 1480 gpm over the YRD (Figure 5c). Associated with the easterly wind anomalies over the northern South China Sea (Figure 5e,f), water vapor transport toward the YRD region is weakened relative to the strong southwesterly monsoonal flow usually prevailing over southeastern China in mid-summer. The drier lower and middle troposphere accompanied by the descending motion leads to smaller cloud cover (<0.4; Figure 5c), which is 0.15 less than the mid-summer average, and consequently positive anomalies of the net short-wave radiation at the surface over the YRD region (>40 W m$^{-2}$; Figure 5f).

In short, during the YRD-only heatwaves, the upper-troposphere jet stream moves northward to the north of 35° N with low wind speeds over the YRD; the WNPSH in the middle and lower troposphere extends westward to the middle and lower reaches of the Yangtze River, leading to descending motion and drier atmosphere over the YRD region. As a result, much fewer clouds are formed and more solar radiation is absorbed at the land surface. The increased solar heating and descending-induced adiabatic heating jointly lead to the high temperature and heatwaves over the YRD region. These results are consistent with previous studies on regional compound hot extremes over the middle and lower reaches of the Yangtze River [33].

### 4.2. The PRD-Only Heatwaves

Compared to the above results, synoptic patterns during the PRD-only regional heatwaves (36 days) differ substantially. At 300 hPa, the GPH over the East China Sea and Taiwan Island is significantly lower than the mid-summer average (Figure 6d), although a closed low-pressure center is not evident in the composite field (Figure 6a). In associated with the negative anomalies of GPH, northwesterly anomalies are observed over the PRD region, which leads to weakening of the tropical easterly jet stream that is often present over the PRD in mid-summer (not shown) and the prevailing northeasterly winds instead (Figure 6a).

The composite fields in the middle and lower troposphere (Figure 6b,c) illustrate a cyclonic circulation centered around Taiwan Island. The low-pressure system centered around Taiwan Island (Figure 6b) is accompanied by negative anomalies of the 500 hPa GPH (up to 30–40 gpm lower than the mid-summer average; Figure 6e) and cyclonic anomalies in the middle and lower troposphere (Figure 6e,f). The prevailing low-level southwesterly winds over the PRD in mid-summer are substantially weakened during the PRD-only heatwaves (Figure 6b,c). The northerly anomalies during the PRD-only regional heatwaves significantly decrease the moisture needed for the production of clouds and precipitation, as the low-level southwesterly wind from the South China Sea is the key provider of water vapor over the PRD region.

The presence of cyclonic circulations in the middle to lower troposphere (Figure 6b,c) suggests a possible impact of TCs on the PRD-only regional heatwaves. A closer examination indicates that TCs are present over the offshore waters and/or the coastal areas of southeast China during 81% (29 out of 36) of the PRD-only heatwave days. The ten TCs (Table 1) mostly move northwestward from the western North Pacific and make landfalling in Fujian Province after passing the Taiwan strait except for Typhoon Nida (1604), Muifa (1109), and Nakri (1412) (Figure 7). Typhoon Nida moves towards the PRD region from the South China Sea, while Typhoon Muifa and Typhoon Nakri move northward to the Yellow Sea without landfalling during the PRD heatwave days. Considering the changes in the synoptic conditions with different locations of the TCs, composite analyses are conducted for four subgroups, respectively, i.e., the 12, 11, 2, and 4 days of the PRD-only regional heatwave accompanied by TCs centered in the four regions (A to D in Figure 7). Note that

the TCs in Region A or B move into Region C or D later, and the TCs centered in Regions A and B are relatively stronger (Figure 7). Except for the four subgroups of the PRD-only heatwave with TCs, the other 7 days do not experience TCs, and the synoptic patterns on these days will be discussed as the fifth subgroup.

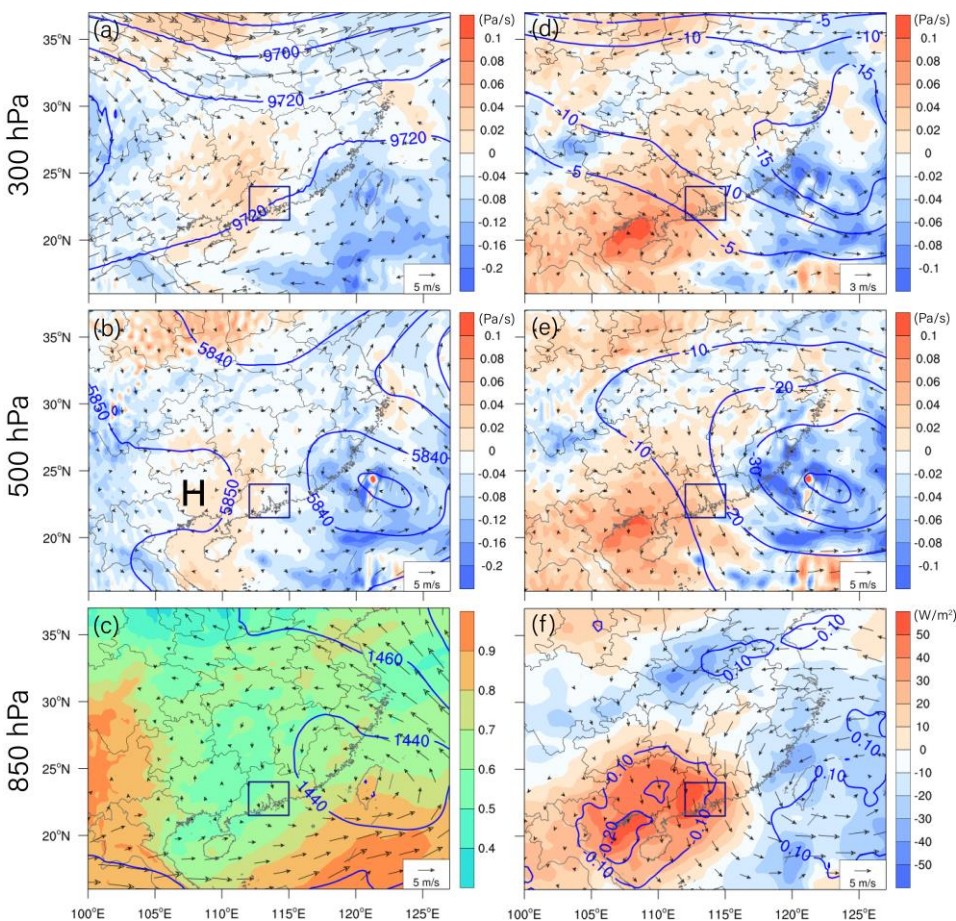

**Figure 6.** Synoptic backgrounds for the PRD-only heatwaves. (**a**) The 300 hPa geopotential height (blue contours; unit: gpm), horizontal wind vector (arrows), and vertical velocity (shadings). (**b**) Same as (**a**), but at 500 hPa. (**c**) Geopotential height (blue contours; unit: gpm) and horizontal wind vector (arrows) at 850 hPa, and total cloud cover (shadings). (**d**) Anomalies of geopotential height (blue contours; unit: gpm), horizontal wind vector (arrows), and vertical velocity (shadings) at 300 hPa. (**e**) Same as (**d**) but at 500 hPa. (**f**) Anomalies of total cloud cover (blue contours), 850 hPa horizontal wind vectors (arrows), and net solar radiation at the surface (shadings; positive downward). Navy rectangle denotes the PRD region. Gray lines indicate the national and provincial boundaries and coastal lines.

The synoptic backgrounds for the five subgroups of the PRD-only heatwave days are shown in Figure 8. With the TCs centered in Region A, the PRD region is situated about 500 km west of the TC center with low-level northerly winds aloft. Note that with the TCs in their mature stage in Region A, the strong upward motion in the cyclonic circulations could force extensively distributed descending to its northwest over the drier inland areas. The PRD region Is located between the ascending and descending regions, where the air vertical velocity seems negligible. Thus, unlike the YRD regional heatwaves induced by prominent high-pressure systems, the synoptic subsidence-induced adiabatic heating is not very strong over the PRD region and the cloud cover over the PRD region is still large (0.7–0.9), suggesting reduced solar radiative heating at the surface. Instead, the heatwaves over the PRD region are closely related to the organized low-level northerly airflows, which transport hotter air from inland mountains to the coastal area of South China (see Figure 1c

for the topography) and the related foehn wind. These relevant physical processes can be seen clearly in Figure 9, which shows the vertical thermal and dynamic structure over the PRD region and the Nanling Mountains to its north during the PRD heatwave days with the TCs in Region A. The near-surface air near the PRD coastline is about 3 K cooler than the northern PRD region, likely due to the large tropical land–sea thermal contrast in mid-summer. Meanwhile, the TCs to the east force northerly winds over South China. When the northerly airflows climb over the Nanling Mountains, the adiabatic cooling leads to the formation of precipitating clouds with liquid water content ($q_c$) of >0.04 g kg$^{-1}$ over the mountains. At the lee side, descending compression causes strong adiabatic heating, as much water vapor has already condensed and been taken away by rainfall. This TC-related foehn effect [16,47,48] increases air temperature at the lee side of the Nanling Mountains, making the northern boundary of the PRD region (at about 24° N) a warm center. This thermal structure combined with the prevailing northerly winds closely associated with the TCs produces intense warm advection over the PRD region, favoring the occurrence of the PRD regional heatwaves. Moreover, the upstream surface sensible heat flux and TC-induced subsidence in its outer circulation could also be contributing factors, as previously documented in a case study [48].

**Table 1.** Basic information about the TCs on the PRD regional heatwave days.

| TC Number | TC's Name | TC's Life Span (mmddhh UTC) |
|-----------|-----------|------------------------------|
| 0808 | Fung-wong | 072400-073118 |
| 0908 | Morakot | 080306-081306 |
| 1109 | Muifa | 072700-080900 |
| 1209 | Saola | 072618-080500 |
| 1410 | Matmo | 071700-072600 |
| 1412 | Nakri | 072900-080400 |
| 1513 | Soudelor | 073000-081206 |
| 1604 | Nida | 072912-080300 |
| 1709 | Nesat | 072500-073100 |
| 1710 | Haitang | 072712-080306 |

When the TCs move from Region A into Region C, the PRD region is located to their south with westerly to southwesterly airflows and weak ascent in the middle and lower troposphere (Figure 8c). The two days of such PRD-only regional heatwave are the last stage of heatwaves induced by TCs in Region A. When the weakened TCs move further northward, the direction of isobaric lines over the PRD region change back to southwest-northeast, and the cooler air brought by the southwesterly winds from the South China Sea rapidly ends the heatwaves.

Compared to the TCs in Region A, the TCs centered in Region B possess similar intensity, but are located several hundred kilometers to the east with the cyclonic circulation hardly extending to the west of Taiwan Island (Figure 8b). Meanwhile, an extensive high-pressure region extends from southeast China to the Yellow Sea (Fee Figure 1a for its location), leading to descending motion and sunny weather. The strong TCs in Region B could contribute to the persistence of the high-pressure system by causing compensating subsidence. Situated roughly at the high-pressure center, the PRD region suffers from heatwaves due to the synoptic subsidence-induced adiabatic heating and enhanced clear-sky solar heating at the surface. The high-pressure system lasts a few days and vanishes after the TCs move from Region B into Region D (Figure 8d). Then, the descending motion over the PRD region is also weakened, while the southwesterly winds over the South China Sea are substantially strengthened (c.f. Figure 8b,d). Consequently, the heatwave over the PRD region is about to end.

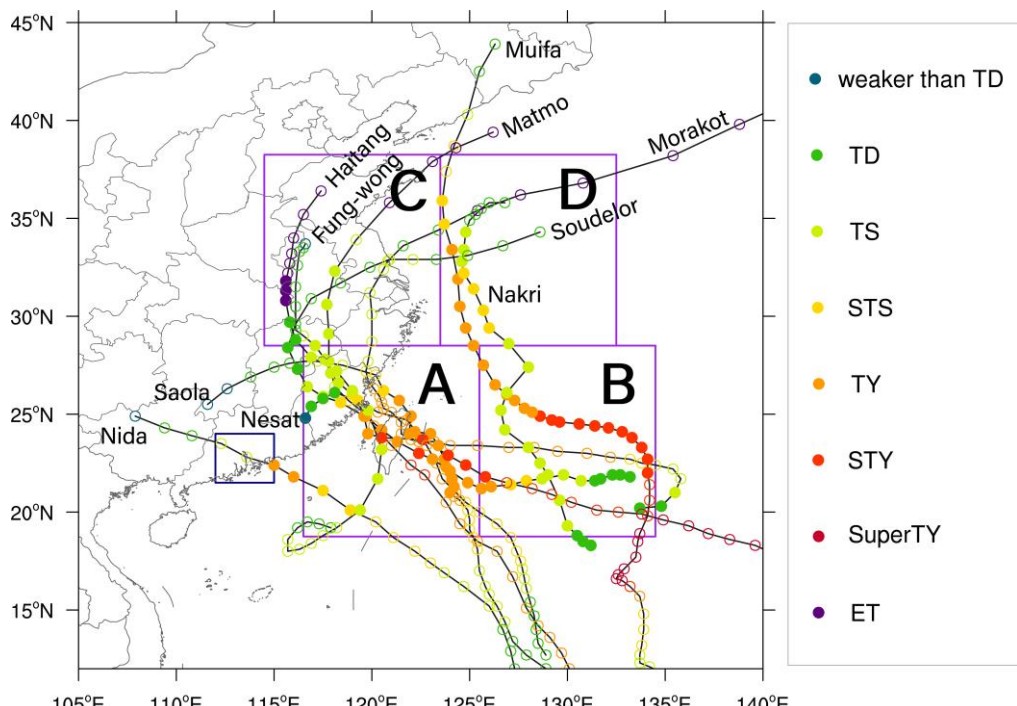

**Figure 7.** Tracks of the ten TCs associated with the PRD-only regional heatwave. The filled and hollow circles denote the TC centers at 6 h intervals on the heatwave days and the other days, respectively, with colors denoting the intensity of TC estimated by CMA: tropical depression (TD), tropical storm (TS), severe tropical storm (STS), typhoon (TY), severe typhoon (STY), super typhoon (SuperTY), and extratropical transition (ET). Purple rectangles denote the four regions where the TCs are centered. Navy rectangle shows the PRD region. Gray lines indicate the national and provincial boundaries and coastlines. A to D represent four regions divided according to different locations of TCs.

The PRD heatwaves without TCs in the western North Pacific is usually associated with a subtropical high over coastal areas of South China and the adjacent water surface (Figure 8e), which explains the descending motion and anomalies over the North Gulf in Figure 6b,e. This subtropical high seems smaller in coverage and weaker in intensity than the strong WNPSH responsible for the YRD heatwaves (Figure 5b), but is comparable to the high-pressure system on the PRD heatwave days with TCs in the Region B (Figure 8b). Under the control of this subtropical high and associated downward motion, heatwaves occur over the PRD region.

*4.3. The YRD and PRD Simultaneous Heatwaves*

Synoptic analysis for the 18 days with simultaneous heatwaves over both the YRD and PRD regions suggests a combination of the major features observed during the PRD-only heatwaves and the YRD-only heatwaves, respectively (Figure 10). Synoptic fields are composited for the 10 days with concurrent TCs and the other 8 days without TC, respectively (Figure 10). As shown in Figure 10a–c, while the WNPSH extends westward to cover the YRD region, the PRD region is situated in the outer circulation of TCs centered several thousand kilometers to the southeast. The dominance of WNPSH over the YRD region is to some extent similar to that during the YRD-only heatwaves (Figure 5a–c) with about the same magnitudes of descent and even smaller cloud cover. On the other hand, the impacts of TCs to the PRD region resemble what occurs on the PRD-only heatwave days with TCs over Region A (Figure 8a) despite that the TCs are relatively displaced southward in the middle troposphere due to the westward extending WNPSH over the YRD (Figure 10b), i.e., the TC-induced foehn effect associated with Nanling Mountains, and possibly the upstream surface sensible heat flux and subsidence induced by the TCs

probably play key roles. Note that about 72% (39 out of 54) of the mid-summer PRD regional heatwave days are accompanied by TCs.

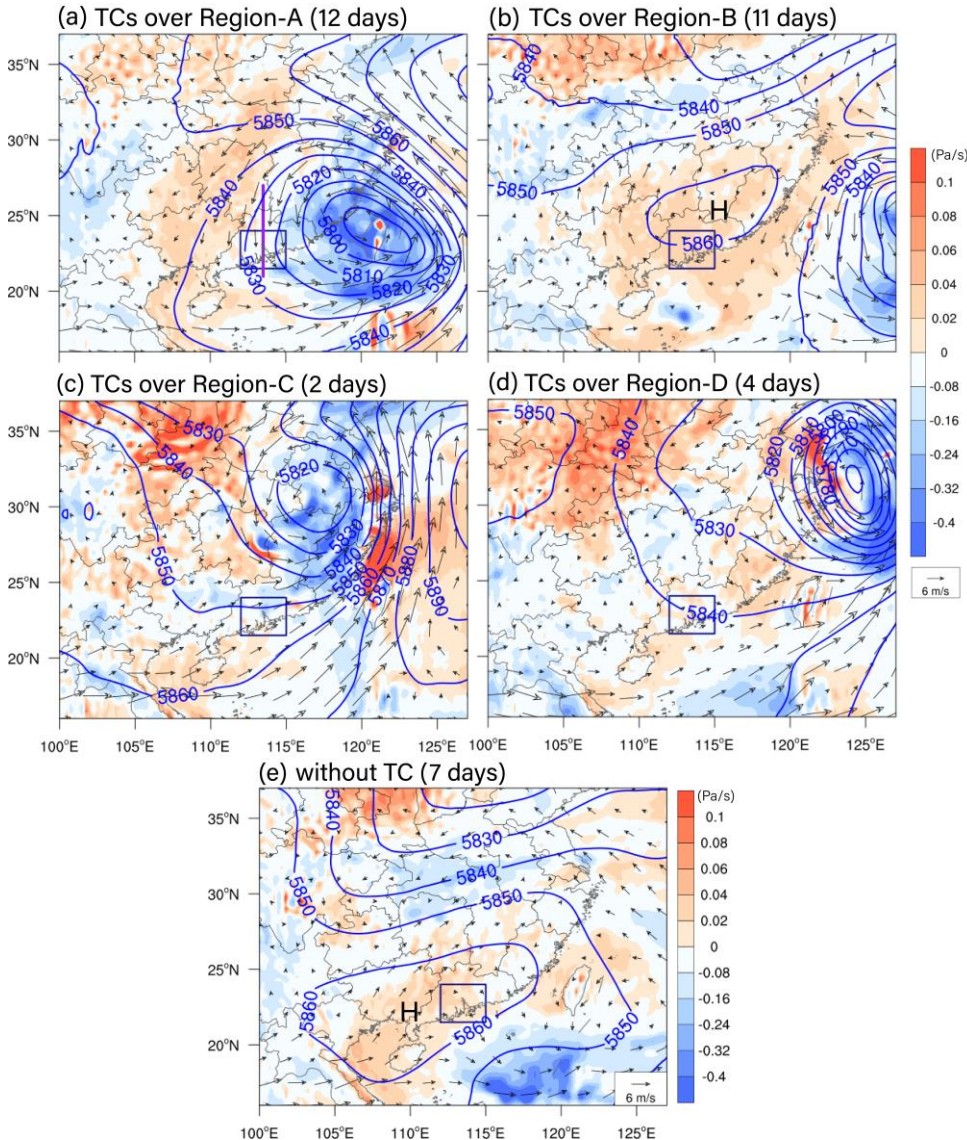

**Figure 8.** Synthetic fields for the five subgroups of the PRD-only heatwave days: (**a–e**) with TCs centered over Regions A, B, C, and D as shown in Figure 7 and I without TC, including 500 hPa geopotential height (blue contours; unit: gpm) and vertical velocity (shadings), and 850 hPa horizontal wind vectors (arrows). Thick purple line in (**a**) represents the location of vertical cross-section used in Figure 9. Rectangular box in each panel denotes the PRD region. Gray lines show the national and provincial boundaries and coastlines.

The other 8 days are characterized by an extremely intense, extensive subtropical high throughout the troposphere, leading to downward motion and few clouds over most areas of southeast China (Figure 10d–f). At 500 hPa, this high-pressure system is the intensified WNPSH and centered roughly between YRD and PRD (Figure 10e). These synoptic features on the 8 days (including 6 days in late July 2016 and 2 days in early August 2017) are like a magnified version of those appearing during the YRD-only heatwave days (Figure 5a–c). They could be related to the weakening of the Somalia cross-equatorial flows and the westerly flow over the tropical Indian Ocean [49].

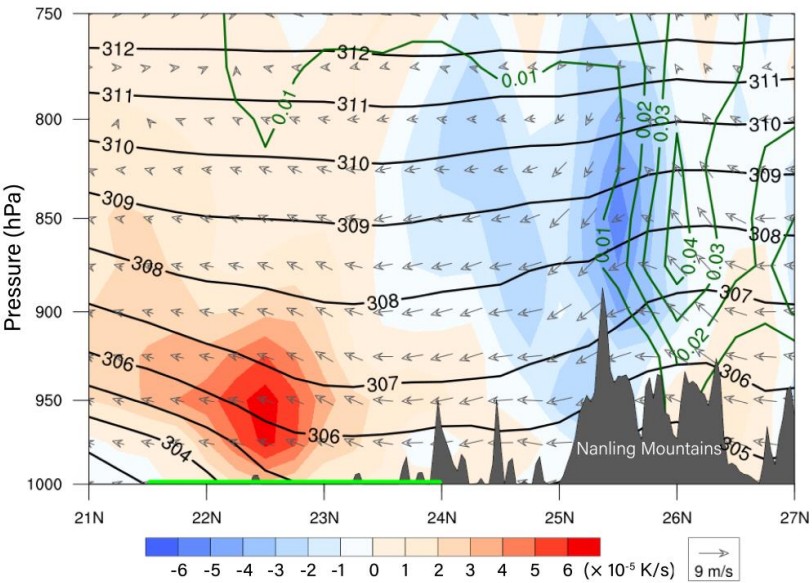

**Figure 9.** Vertical cross-section along the thick purple line in Figure 8a for the PRD-only heatwave days with a TC centered in Region A. Arrows denote the wind vectors along the plane. The vertical velocity is multiplied by a factor of 200. Black contours denote the potential temperature ($\theta$; unit: K). Green contours denote the mixing ratio of cloud liquid water ($q_c$; unit: g kg$^{-1}$). Shadings denote the horizontal advection of $\theta$. Gray shading denotes topography. Thick green line indicates the location of the PRD region.

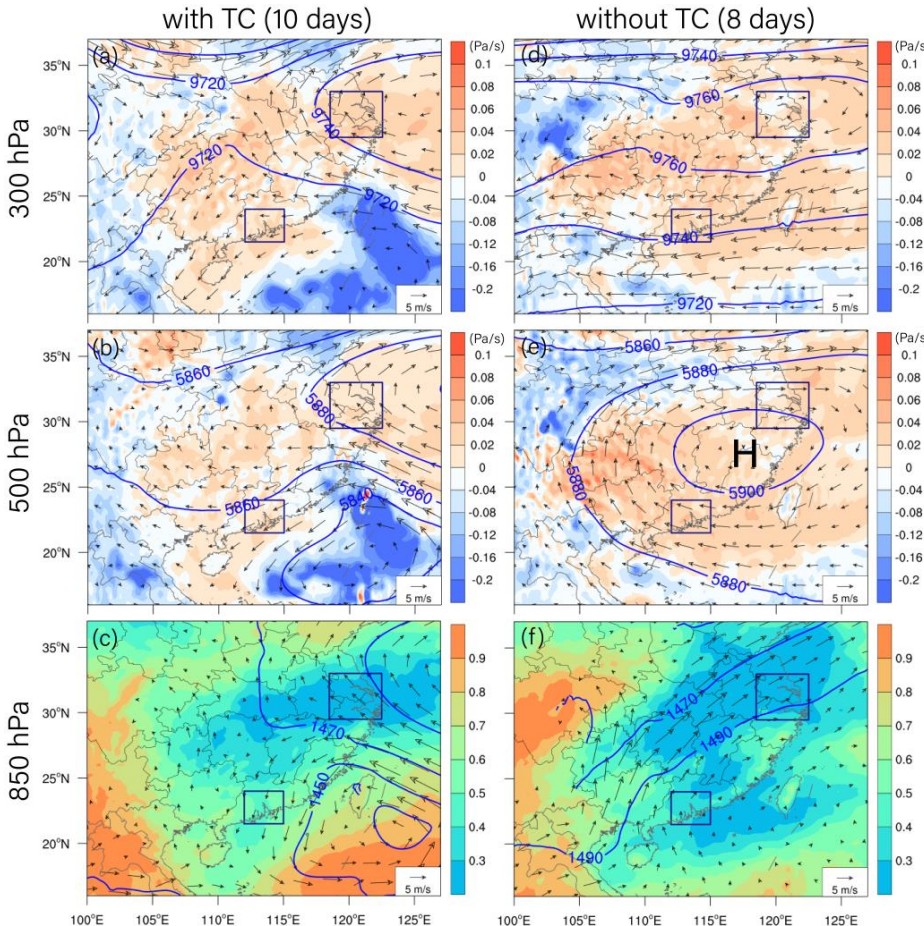

**Figure 10.** Synoptic backgrounds for the heatwaves simultaneously over both YRD and PRD regions

(rectangular boxes), with TC (left) and without TC (right), respectively. Top panels (**a,d**) represent the 300 hPa GPT (blue contours; unit: gpm), horizontal wind vector (arrows), and vertical velocity (shadings; positive and negative value corresponds to descending and ascending motion). Middle panels (**b,e**) are same as the top panels except for at 500 hPa. Bottom panels (**c,f**) represent the 850 hPa GPH (blue contours; unit: gpm) and horizontal wind vector (arrows), and total cloud cover (shadings). Gray lines show the national and provincial boundaries and coastlines.

## 5. Summary and Conclusions

EHEs exert adverse effects on human health, the economy, and the ecosystem, and have increased with climate warming and global urbanization. Given the large population exposed to the high temperatures in big cities, it is important to deepen the understanding of the characteristics and mechanisms of urban EHEs.

The YRD and PRD in the coastal areas of southeast China are the most flourishing economic areas in East Asia that suffer from EHEs in boreal summer. Using the observation of dense surface weather stations and high-resolution reanalysis data, this study comparatively analyzes EHEs over the two urban agglomerations in ten warm seasons (May to August 2008–2017). Unified metrics are used to define and characterize heatwaves and warm nights at the meteorological stations in both regions, and their spatial–temporal distributions are revealed. Moreover, the regional heatwave of each region is defined based on the proportion (>0.5) of stations simultaneously going through a heatwave in the region. Synoptic backgrounds for the regional heatwaves over the YRD only, PRD only, and both the YRD and PRD simultaneously are discussed in detail. The major conclusions are as follows.

(1) Both regions observe substantial spatial variations of heatwaves, reflecting the modulation of the local geographic environment on the characteristics of heatwaves. The inland mountains experience the most frequent heatwaves, while the coasts experience the least due to the cold advection of sea breeze. Despite little difference in heatwave frequency between urban and rural areas with similar geographic environments, cities suffer from much more frequent and intense warm nights in heatwaves than the rural areas in both the YRD and PRD.

(2) All the regional heatwaves over the YRD and PRD regions occur in summer (JJA). Those over the YRD are more concentrated in mid-summer (from 15 July to 15 August), while those over the PRD start earlier in June and end later at the end of August. This partially explains the more intense heatwaves in the YRD region with a warm-season average cumulative intensity about 3 times that in the PRD region.

(3) The persistent, enhanced WNPSH in the middle to lower troposphere is the dominant system responsible for the mid-summer YRD heatwaves, which account for about 80% of the YRD heatwaves in the ten warm seasons. The strong WNPSH extends over the mid-lower reaches of the Yangtze River, forcing a significant descending motion. Meanwhile, the water vapor transport to the YRD region is weakened, as the prevailing southwesterly airflows over South China are replaced by southeasterlies. Consequently, much more solar radiation (>40 W m$^{-2}$) is absorbed at the land surface than the mid-summer average. The descending adiabatic heating and enhanced solar heating jointly produce heatwaves. These are generally consistent with previous studies [33,50].

(4) About 53% of the PRD heatwave days are in mid-summer and influenced by both TCs and subtropical high-pressure systems. Among those days, about 72% experience TCs to the east of the PRD region. With the TCs centered around Taiwan Island, organized low-level northerly airflows prevail over the PRD and inland areas to its north. The warm air climbs over the Nanling Mountains and is heated at the lee side by the foehn effect and likely the surface sensible heat flux, and further advected to the PRD region. On the PRD heatwave days with a remote TC to the east of Taiwan Island or without TCs, a local high-pressure system is the major driver.

(5) Regional heatwaves could occur simultaneously over both the YRD and PRD under two types of synoptic configurations. One is characterized by an extremely intense WNPSH covering southeast China; the other features a coupling between an enhanced WNPSH extending the over YRD and TCs centered to the south of Taiwan Island causing northerly low-level airflows over the PRD and the Nanling Mountains to its north.

This study clearly demonstrates that urbanization has increased the occurrence of warm nights over the two major urban agglomerations in southeast China, which could aggravate the detrimental impacts of heatwaves. This result is consistent with the previous findings of Chen and Zhai [43] and Gao et al. [44] and calls on the government, enterprises, and the public to take joint action to reduce the urban heat island effect. Possible measures include planting more green plants in urban areas, producing and using more energy-efficient air conditioners and cars, and building residential buildings with good ventilation and heat dissipation functions. Moreover, the statistical analysis in this study quantitatively reveals the contributions of both TCs and high-pressure systems to the mid-summer EHEs over the PRD, and supplements the previous finding about the role of the TC-induced foehn effect in the occurrences of EHEs in the downslope regions of Taiwan [16] and southeastern coast of China [48]. As TCs are projected to be more intense with global warming [49,50], the adverse impacts of TC-induced EHEs on global society might increase [51]. However, changes in EHEs over the PRD region and other cities worldwide in the future are unknown and deserve further study, as such regional EHEs depend on the track and movement of TCs, whose projection is associated with large uncertainties.

We should acknowledge the limitations of this study and point out that more investigations are needed in the future. The number of EHEs can be increased by extending the analysis period to improve the statistical significance and robustness of the conclusions. For the PRD heatwaves accompanied by TCs centered around Taiwan Island, the possible role played by the upstream surface sensible heat flux is speculated based on a previous study for an extreme heat event induced by Typhoon Lekima [48], and quantitative analysis for our cases is needed. Further investigation on the duration of the heatwaves under different circulation patterns, predictability of the circulation patterns, and their linkage with atmospheric variability modes and sea surface temperature is important and should help the prediction of the EHEs in the PRD and YRD regions. Furthermore, using the thermal biometeorological indices widely used in thermal discomfort assessment such as PET and UTCI [52] would supplement the characteristics of EHEs based on temperature only.

**Author Contributions:** Conceptualization, Y.L., X.G. and X.S.; methodology, Y.L., X.G. and X.S.; software, X.S.; validation, Y.L., X.G. and X.S.; formal analysis, X.S.; investigation, X.S. and X.G.; resources, Y.L. and X.S.; data curation, X.S.; writing—original draft preparation, X.G. and X.S.; writing—review and editing, Y.L., X.G., X.S., W.-K.W. and H.X.; visualization, X.S.; supervision, X.G. and Y.L.; project administration, Y.L.; funding acquisition, Y.L. All authors have read and agreed to the published version of the manuscript.

**Funding:** This research was funded by the Guangdong Major Project of Basic and Applied Basic Research (2020B0301030004) and the National Natural Science Foundation of China, grant numbers 42030610 and 41975106.

**Data Availability Statement:** The ERA5 Reanalysis dataset is available at https://cds.climate.copernicus.eu/cdsapp#!/home (accessed on 5 September 2022); DMSP/OLS nighttime lights data are available at https://www.noaa.gov/ (accessed on 10 June 2022); the TC best track dataset is available at https://tcdata.typhoon.org.cn/ (accessed on 30 October 2022). Temperature data were provided by the National Meteorological Information Center and the Hong Kong Observatory, which are available from the authors upon reasonable request.

**Conflicts of Interest:** The authors declare no conflict of interest.

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
