# Peer review of "A Comparative Analysis of Characteristics and Synoptic Backgrounds of Extreme Heat Events over Two Urban Agglomerations in Southeast China"

_land, doi:10.3390/land11122235_

Round 1

Reviewer 1 Report

The manuscript analyzed the synoptic features leading to heatwaves over two regions of the Southeast China agglomerations, as well estimated the urban features effect on heatwaves characteristics. Generally, the scope of study is of course topical, there are significant results obtained including typical circulation patterns analysis. The study brings a bit of the novel knowledge to regional heatwaves formation processes. However, there are some questions and remarks listed below.

1.    More relevant information and background should be presented in the Introduction Section about the Urban Heat Island effect, specifically for the considered regions. Since the UHI is very important feature affecting heatwaves, it is important to provide an overview of current knowledge about statistics, magnitude and other characteristics of this feature focused on the study region and the specific largest cities. It is also important because Authors explained some features in text relying on the UHI features in these regions without references.

2.    Authors considered 2008 – 2017 period. It should be indicated, why the such short period was considered? This leads to another question about relevance of 10-years period to get statistically significant and robust results. Does it enough to get justified conclusions?

3.    Authors applied the term “warm nights” defining it as days Tmin > 27 0C (95th percentile of Tmin). However, there is well-known term “tropical nights” defined as days with Tmin > 20 0C. Could you please explain, why the “warm nights” was selected instead of “tropical nights”?

4.    There are many thermal biometeorological indices widely used in thermal discomfort assessment during heatwaves: PET, UTCI, etc. Perhaps, using these indices would get more reliable results in heatwaves estimations, and would be useful to be compared to simple temperature characteristics.

5.    Authors analyzed two distinct regions: PRD and YRD during the warm season as May – September. However, the Pearl River region is situated more closely to the equator and strongly affected by monsoons during June. Therefore, the warm season there could started from April or, maybe, even in the last March. Maybe, it would be useful to include April into consideration in PRD, or to provide climatology justified there are no heatwaves observed.

6.    Authors used the same threshold to determine heatwaves days, according to 95th percentile of Tmax in considered regions. However, in my opinion, 95th percentiles of Tmax could differ significantly as between PRD and YRD, as well inside of each region. Please, provide statistics of 95th percentiles of Tmax for considered regions, as well its variability inside each region to justify the choice of 35 0C as the unified threshold for both domains. The same question concerned to Tmin threshold.

7.    Authors provided some statistics about spatially averaged warm nights and EHEs characteristics for both regions. It would be more clarified to be presented as Table including a bit more statistics (e.g., not only average values, but also its variability).

8.    It would be interesting to plot the yearly frequency of heatwaves within each region from 2008 to 2017 (including frequencies according to Authors’ heatwaves’ classification).

9.    Anomalies in Figures 5 etc. are calculated with respect to the mid-summer. What does mean “mid-summer” – which period? Please clarify in text. It leads also to question, how relevant is this mean value for a few-days process, such as heatwave?

10.    The question about figures 6a and 6d. With respect to the tropical easterly jet, the composite pattern differs from it by northeasterly winds; however, the anomalies are northwesterly; this is inconsistence, or my misunderstanding; therefore, please, clarify this in text.

11.    Please evaluate the significance of difference between Regions- A, B, C and D samples. Are these samples differing in statistical sense, are these samples enough robust? The same question is for TC and not-TC samples, as well for simultaneous heatwaves sample and PRD-only and YRD-only samples.

12.    Does Figure 9 presenting the composite pattern for Region A PRD-only heatwaves? If so, please indicate it in text.

13.    Figure 10 has poor readability, please increase the image’s resolution.

14.    Authors indicated the sensible heat fluxes as important favorable condition (among others) in heatwave formation, however there are no any quantity analysis presented. Please provide it using the ERA5 data, for example.

15.    The important question is how long the assigned circulation patterns are forming, and how long they are lasting to get the heatwave effect? It would be useful to provide some statistics about the duration of heatwaves according to different circulation patterns. It is also the question in context of rapid transition of TC, therefore does TC have enough time to form heatwave effect during its passage according to the assigned circulation type patterns?

16.    The Discussion section should be moderately extended. There is just no context of comparison with other results and previous works. How the suggested explanations are consistent with other results?

17.    Conclusion should include also the perspectives discussion. One of the most important appears the importance of predictability of assigned circulation patterns, as well its linkage with atmospheric variability modes relevant for considered regions: such as TBO, IOD, PNA, PDO, ENSO, etc. This could improve the predictability of heatwaves in considered regions.

The most important questions concerned more to methodological issues, extensions and explanations, which are needed to be answered and/or included in text. However, these issues are not crucial for results, therefore, I recommend this manuscript to minor revision.

Reviewer 2 Report

Comments to "A comparative analysis of characteristics and synoptic backgrounds of extreme heat events over two urban agglomerations in Southeast China" by Sun et al.

This paper presents systematic analyses regarding the spatiotemporal characteristics and synoptic backgrounds of extreme heat events over YRD and PRD. The paper is well written. I suggest the possible publication in Land if the following concerns are addressed.

1. Are there any differences in spatiotemporal characteristics and synoptic backgrounds for extreme heat events over YRD and PRD, if the authors employ relative temperature methodology (i.e., Tmax > 90th percentile) to define a heatwave day?

2. Further discussions can be added to show and elucidate the different sea surface temperature backgrounds concerning YRD-only heatwaves, PRD-only heatwaves, and YRD and PRD simultaneous heatwaves. This may promote the forecasts of heatwaves.

Minor comments:

1. Names of TCs should be given in Fig. 7.

2. The method of how DMSP/OLS nighttime lights data identify the urban regions should be given in Section 2.

Reviewer 3 Report

Thank you for the opportunity to review this interesting paper. The case for the study and research gap are clearly identified. Urban Heat Island is well established, I think it would be of benefit to emphasis what you are adding to that discussion - perhaps in section three. Some of the figures presented are lacking some sharpness, higher resolution would be of benefit. The titles of the figures are too long, instead of giving an explanation of what the figure says, I would prefer just the title and then an explanation about the figure in the text. The paper gives a good explanation of what the existing situation is, based on an appraisal of existing data. I think it would be of benefit if the implications of this could be drawn out in more detail - for instance what does this mean for urban planning, the built environment, residents, health, the future (with the potential impacts of climate change)? what should be done and by whom to address these issues?
